# Gender Identity Orientation and Sexual Activity—A Survey among Transgender and Gender-Diverse (TGD) Individuals in Norway

**DOI:** 10.3390/healthcare12040482

**Published:** 2024-02-16

**Authors:** Elsa Mari Almås, Esben Esther Pirelli Benestad, Silje-Håvard Bolstad, Tor-Ivar Karlsen, Alain Giami

**Affiliations:** 1Department of Psychosocial Health, University of Agder, 4604 Kristiansand, Norway; esben.esther@uia.no (E.E.P.B.); silje.havard@uia.no (S.-H.B.); tor-ivar.karlsen@uia.no (T.-I.K.); 2INSERM (National Institute of Health and Medical Research), 94807 Paris, France; alain.giami@inserm.fr

**Keywords:** transgender, gender-diverse, gender identity, sexuality, gender diversity, gender orientation

## Abstract

Background: The understanding and conceptualizing of gender and sexuality are continuously negotiated between individuals and cultures. Recently, new gender identity orientations have emerged, fighting pathologization and establishing new spaces and options for being sexually active gendered beings. Objective: To investigate variations in sexual activities across different gender identity orientations. Method: A questionnaire used in France was adapted to the Norwegian context and implemented in this study. The participants were recruited through therapists, TGD organizations, and social media. Results: A total of 538 individuals responded to the questionnaire, of which 336 provided a written description of their gender identity. Based on an analysis of the degree of male gender identity orientation, the degree of female gender identity orientation, and the degree of nonbinary gender identity orientation, three clusters appeared and were used in the analyses of sexual activities and preferences. Conclusions: Some findings could be attributed to lingering aspects of traditional gender roles, while others may be indicative of sexual expression stemming from societal acceptance of gender diversity and new identity orientations.

## 1. Introduction

### 1.1. Historical Background

Since sexologists at the beginning of the twentieth century began to describe sexual and gender diversity as presented by their clients, there have been competing approaches for understanding and conceptualizing these phenomena. The initial accounts of sex and gender nonconformity by medical practitioners—von Krafft-Ebing in the late 1800s [1] and Hirschfeld in 1910 [2], in addition to Freud in 1905 with his psychoanalytic theories [3]—brought gender diversity into the purview of psychiatrists. There was a prevailing notion that sex/gender identity and expression that did not align with the assigned sex at birth constituted a form of deviance [4].

Despite an increasing comprehension among professionals of the intricacies of sex and gender, the binary model was the prevailing paradigm throughout the twentieth century. The development of the two-sex model, which evolved from an earlier one-sex model, was elaborated by Thomas Laqueur [5].

In English, there used to be no word to elucidate the difference between sex, defined as somatic characteristics, with the genitals as the primal signifier, and the subjective experience of having a male or female gender identity. In 1952, John Money put forth the concept that one’s gender role was shaped by personal experiences with having either male or female reproductive organs and the manner in which the parents raised the child in alignment with these characteristics [6].

Gender identity has been understood as the subjective experience of being man or woman [7], both, or neither [8,9,10]. In a dimensional understanding, gender identity can be understood as a normal expression of neurobiological variation [11]. A categorical understanding is usually based on the traditional conviction that there are two—and only two—genders based on the reproductive sex organs, and deviation from gender identity in congruence with the sex organs is regarded as pathological [12]. Pathologization has given rise to much shame and inner conflict among trans individuals who have been subject to years of unsuccessful and harmful attempts to treat what was regarded as a delusion. Sexuality was a subject that trans people understood that they should not talk about if they wanted medical body adjustment.

The pathologizing of transsexuality was first openly challenged by the American doctor Harry Benjamin: “Since it is evident, therefore, that the mind of the transsexual cannot be adjusted to the body, it is logical and justifiable to attempt the opposite. To adjust the body to the mind” [13] (p. 53). He also wrote: “For the simple man in the street, there are only two sexes. A person is either male or female, Adam or Eve. With more learning comes more doubt” [13] (p. 6).

John Money’s theory on gender identity, as outlined in Money and Ehrhardt’s work from 1972 [7], faced a significant challenge in the 1990s because of the reevaluation of the John/Joan case. This case involved a baby boy who, in the 1960s, experienced a penile injury during circumcision, leading to subsequent surgery to construct female genitalia, in line with John Money’s theory, resulting in John having to transition to Joan. This was, for many years, seen as a success story, but in 1997, a new publication portrayed the whole story as a traumatic experience for John/Joan. When informed about his background at the age of 14, he requested reoperation to align with the gender he identified with [14,15].

Following the revelations from the John/Joan case, an expert committee was formed, and in 2006, they published a report, a key conclusion of which was that gender identity is a neurobiological variation, not a mental disorder [11].

The understanding of gender roles as shaped predominantly by environmental influences faced challenges in the post-2000 era because of new research on epigenetics and a growing recognition that the mind and body are more interconnected than previously believed [16,17]. Transgender individuals often express profound internal longing, describing a sense of incongruence between their gender identity and physical bodies [18].

Several researchers have highlighted the significant diversity inherent in biological sex, as exemplified by individuals who have received intersex diagnoses [11,19,20,21]. Understanding gender identity as a neurobiological variation raises new questions, as Ann Fausto-Sterling implied in the title of one of her papers: Gender/Sex, Sexual Orientation, and Identity Are in the Body: How Did They Get There? [8].

We do not know the final answer, but it has been suggested that there are aspects of gender identification that may be based on our genome [18]. Brain research has also pointed in the same direction, indicating that rather than being merely shifted toward either end of the male–female spectrum, transgender persons seem to present with their own unique brain phenotype [22,23,24].

The WHO confirmed the depathologization of gender incongruence in the ICD-11 Classification of Mental and Behavioral Disorders. A new chapter, Sexual Health, was introduced to include the diagnosis of gender incongruence (HA60, 61, and HZ), thereby removing it from the chapter on mental disorders. The revision of the ICD-11 has been paralleled by the development of a higher degree of self-affirmation in the trans communities and by trans individuals entering the professional scene and introducing trans subjective perspectives [25].

### 1.2. Gender Identity Orientations among Transgender and Gender-Diverse People

The capacity for bodily modification through medical procedures may have played a crucial role in shaping the self-perception of transsexual individuals [26,27,28] and could have reinforced the emphasis on genitalia in defining gender [27,29,30], here based on a binary understanding of sex. The option for body adjustments through hormones and surgery has allowed transsexual individuals to align their physical selves with their subjective gender identity as they strive to become what they feel they truly are: either men or women.

Dallas Denny and Cathy Pittman described the term “transgender” as having originated from gender-variant individuals during the 1990s. The word “transgender” has evolved into a term that encompasses a diverse group of individuals whose gender identities and expressions challenge the conventional binary norms. Kate Bornstein considered the term “transgender” as short for transgressively gendered. This umbrella term covers all gender-variant people, including transsexuals [31,32]. The concept of “transgender and gender-diverse” (TGD) is not limited to representing a specific category or group; rather, it allows for the recognition of various potential gender identity orientations (GIOs) across new dimensions [33].

Currently, there are numerous alternative perspectives on understanding gender identity [34,35,36]. In the 1970s, Sandra Bem conceptualized femininity and masculinity as two independent dimensions New possibilities opened up, allowing individuals to embrace both masculine and feminine qualities and neither simultaneously [37,38].

The concept of a nonbinary notion of transgenderism appears to be relatively recent. A search conducted in library databases (the search was performed on 15 November 2023), including Medline, Cinahl, SocIndex, Teacher Reference Center, Health and Psychosocial Instruments, and APA PsychInfo, using the terms “nonbinary” or “nonbinary” in the title and “gender” in the abstract, revealed two peer-reviewed publications before 2010 and a sharp increase in publications from 2015 (n = 5) to 2020 (n = 120) and then until the end of 2022 (n = 282). This surge in publications reflects a growing professional interest in GIOs within the transgender community, especially in nonbinary GIOs.

The concept of GIO allows TGD and other individuals to be positioned within three different dimensions: degree of maleness, degree of femaleness, and a dimension from binary to nonbinary. Although numerous challenges persist for TGD individuals, progress has been made toward reducing pathologization and increasing acceptance, creating a self-defined space where one can embrace both gender identity and sexuality.

### 1.3. Sexuality among Trans and Gender-Diverse People

Historically, sexuality among TGD individuals has been overlooked or intentionally excluded in much of the literature. This omission was based on the assumption that individuals seeking sexual reassignment surgery (SRS) did not have an active sex life, leading to the belief that they were candidates for surgery, even if it might diminish their erotic sensitivity and sexual function [27]. It was widely accepted that trans-sexuals desiring SRS did not experience sexual feelings or needs. However, this once perceived “truth” has been shown to be incorrect [39,40,41,42,43,44,45].

In 2015, Sari van Anders introduced the concept of gender/sex as an identity that encompasses both biological and social aspects intertwined with sexuality within what she referred to as a sexual configuration. In this framework, sexuality can be viewed as an integral part of a social context and recognized as a complex and fluid aspect of human identity that evolves over time. Van Anders’ ideas align with the observations made by Hirschfeld at the start of the twentieth century [3], emphasizing the interconnectedness of sexual and gender identities. Sexual attraction can involve sex/gender, age, status, norms, number of partners, type of sexual activity, intensity, single sex, and consent. Sexual orientation can involve sexual wishes and sexual needs, social roles, affirmation, assurance, safety, and being comfortable. Sexual identity can involve gender/sex sexualities (sexualities among individuals who experience gender incongruence, homo-curiosity, gender-independent sexuality, ungendered sex) and normative, lesbian, gay, BDSM, fetish, polyamorous, asexual, player, monoamorous, and kink identities [3,46].

Anne Fausto-Sterling presented a tradition where gender/sex, sexual orientation, and identity are viewed as elements of a larger system. Fausto-Sterling introduced the orthogonal turn, a departure from traditional binary oppositions, such as social versus nonsocial, or nature versus nurture. Instead, this approach integrates and interweaves aspects of sex, gender, sexual orientation, bodies, and cultures without imposing a requirement to select one over the other [8].

As per the sexual configuration theory by van Anders and the orthogonal turn explicated by Fausto-Sterling, an individual can have a body that can be classified as a male body, here based on appearance, without knowledge about the biological internal configuration. This individual can, as a case in point, have a female gender identity, experience a male body consciousness, present a male but androgyne body picture, have a feminine gender role, and be attracted to lesbian women [47].

In conjunction with the growth of self-affirmation and pride within the transgender community, there has been a noticeable increase in the number of individuals seeking counseling and treatment from gender clinics [41,48,49,50,51]. Many of these clients are challenging the traditional binary model of gender and are exploring positions beyond conventional gender norms [19,52,53,54]. Some have even embraced a degendering approach to their bodily sex characteristics. When working with transgender individuals in counseling and therapy, it is apparent that there is greater diversity in what is considered sufficient in terms of bodily adjustments. Some individuals have sought treatment to achieve a fully male or female body, while an increasing number have opted not to undergo complete SRS or are interested in partial SRS [55]. The present study aims to investigate variations in sexual activities across different GIOs.

## 2. Methods

The research presented in the current study is grounded in data gathered from Norwegian transgender individuals and incorporates clinical perspectives on their sexual activities. Alain Giami, Emmanuelle Beaubatie, and Jonas Le Bail collaborated with transgender organizations and healthcare experts in France to create a survey questionnaire [55,56]; this questionnaire featured one open-ended question designed to capture gender identities among the participants by asking the respondents to describe their current gender identification using their own phrasing. The same protocol was developed in Brazil [57], Italy [58], Chile [59], and Portugal [60].

The French questionnaire [55,56] was translated and culturally adapted to fit the Norwegian context by utilizing a forward–backward translation procedure in which the French questionnaire was translated to Norwegian by a bilingual translator [61]. Cultural differences between Norway and France were discussed in the Norwegian research team, thus requiring a few changes in the questionnaire. Finally, the questionnaire was backtranslated to French, and the French version was reviewed by the French authors before its use in the survey. The final version of the questionnaire was tested with a reference group of transgender individuals (n = 10).

Partnerships were forged with TGD organizations and healthcare professionals. The study was overseen by a steering committee that included transgender representatives. The participants were recruited through activist organizations and therapists who worked with TGD clients. Three TGD organizations agreed to distribute the questionnaire among their members, and five centers that regularly provided counseling and support to TGD individuals agreed to distribute the questionnaire to their clients.

The questionnaire comprised 129 questions that were organized into six sections: (1) sociodemographic variables; (2) details regarding the transition process and medical and psychological care; (3) health and sexual health, encompassing topics such as HIV and sexually transmitted infections; (4) mental health; (5) aspects of sexuality, including sexual behavior, sexual difficulties, and attitudes toward sexuality; (6) closing questions related to gender identity and experiences of discrimination. The present paper focuses specifically on the variables related to sexual behavior and GIO.

The questionnaire was administered online through a web page hosted by the University of Agder, Norway. An invitation letter was sent to the partners, and a separate letter was addressed to the participants. These letters were accompanied by a hyperlink to access the questionnaire. Individuals interested in participating, whether affiliated with organizations or clinics or who discovered the survey via social media, were encouraged to visit the website and complete the survey. The online questionnaire was available from 4 April to 1 August 2018. We did not collect or record email addresses or any other digital tracking information.

The survey participants were given the following question: “Regarding gender identity, how do you define yourself right now (describe in text)?” A total of 336 respondents provided answers to the question. Among these, many were identical (e.g., “man”; “nonbinary”), which is why the list of 336 self-descriptions was condensed into a list of 193 unique self-descriptions. A panel of 20 expert raters was assembled through direct contact to conduct a qualitative analysis of the self-descriptions. Half of these raters (n = 10) were experienced clinicians or researchers in the field of transgender health; the other half were transgender community representatives with comprehensive knowledge gained through their personal experiences and extensive connections within the transgender community.

The raters underwent training for their tasks, including written material and a video meeting. They were introduced to a triple-scale model of gender identity that was developed specifically for the present study. This model encompassed three separate scales representing the degree of femaleness, the degree of maleness, and the degree of binaryness in gender identity. These three dimensions were considered distinct and not necessarily interrelated.

Subsequently, the raters individually assessed the 193 unique self-descriptions of gender identity one at a time. Their task was to gauge the levels of femaleness, maleness, and binaryness associated with each self-description. The raters performed this task independently on their own computers, and the time required to complete the task varied from approximately 30 to 90 min. The tasks and responses were administered through a dedicated website designed for this purpose.

The inspection of the 20 independent sets of ratings indicated considerable divergence between the raters. To address this, a composite set of the three scales (femaleness, maleness, and binaryness) was created by averaging the responses from all 20 individual raters on a scale from 0 to 1. For instance, in the case of the unique self-description “nonbinary transmasculine”, the final binaryness score for that specific self-description was derived from the average score on the binaryness scale.

Subsequently, a K-means cluster analysis was conducted, resulting in the identification of three distinct clusters. These clusters were labeled “female gender identity orientation” (FGIO), with a sample size of 93, “male gender identity orientation” (MGIO), with 148 respondents, and “nonbinary gender identity orientation” (NBGIO), with 93 respondents.

In addition to the cluster analysis, other statistical analyses were carried out in the present study, including cross-tabulations with Pearson’s chi-square testing and Kruskal–Wallis tests. The predetermined threshold for statistical significance was established at *p* < 0.05.

## 3. Results

The sample in the current study cannot be regarded as representative of the trans population. There have also been divergent data on the examined parameters in different epidemiological studies [51]. When it comes to age distribution, almost two-thirds of the respondents in the present study were under 30 years old. As seen in Table 1, most of those who oriented themselves toward an MGIO (MGIO: male gender identity orientation) were younger than those who oriented themselves toward an FGIO (FGIO: female gender identity orientation) or an NBGIO (NBGIO: nonbinary gender identity orientation). A larger proportion of the MGIO respondents had less formal education than the FGIO and NBGIO respondents.

Table 2 demonstrates that a significantly higher proportion of MGIO individuals reported sexual satisfaction in terms of seducing and dominating their partners compared with both FGIO and NBGIO respondents. Conversely, fewer MGIO respondents reported sexual satisfaction when it came to using clothing and other fetishes. In contrast, a larger percentage of FGIO respondents reported sexual satisfaction “when giving themselves to someone”.

The MGIO respondents (Table 3) had the earliest sexual debut with another person, while all groups were equal in their sexual debut with themselves around 12–13 years of age.

All groups were turned on by both men and women, whereas the NBGIO group was more inclined to be pansexual than the other groups (Table 4).

As shown in Table 5, among the groups, the FGIO individuals were more prone to experiencing sexual arousal when perceiving themselves as a woman. The MGIO group, on the other hand, reported being more sexually aroused by male clothing, engaging in male-associated activities, and identifying themselves as a man compared with the other groups. The NBGIO group tended to be more sexually aroused by the concept of seeing themselves as transgender. Autoeroticism, or self-pleasure, was found to be common among the respondents from all groups. Notably, all the items in the table were identified as sources of sexual interest by the members of each group.

The FGIO and MGIO groups were both more likely to have had their most recent sexual experiences with a woman (Table 6). In contrast, the nonbinary GIO group displayed a more even distribution, with their most recent sexual experiences involving a man, a woman, or a transperson.

The FGIO respondents expressed greater interest in activities such as kissing, vaginal penetration, and oral sex compared with the other groups. They also engaged in a higher number of sexual activities overall (Table 7). Conversely, when it comes to the frequency of masturbation in the past 12 months, the FGOI respondents had the lowest rate compared with both MGIO and NBGIO respondents.

## 4. Discussion

The primary objective of the current study was to investigate the sexual activities of TGD individuals in relation to their GIO. These orientations must be regarded as dimensions, not as categories, in the interpretation of the results. The findings from this survey underscore the wide range of gender identifications within the transgender community. From the self-descriptions provided by the respondents, we identified trans identities as differentiated, which can be better understood as dimensions related to the degree of maleness, femaleness, and nonbinaryness rather than polarizations, in line with Sandra Bems and Anne Fausto-Sterlings’ suggestions. As a new development, we observed more bisexuality and sexual activities that can be further described as configurations, as suggested by Sari van Anders.

It is now widely recognized that transgender individuals are sexually active [41,42]. The respondents to this survey reported being sexually active in all aspects relevant to the research. A significant portion of the respondents considered sexuality to be crucial for their overall well-being. However, less than half of the respondents expressed satisfaction with their sexual lives.

Our findings indicate that, to some extent, the sexual preferences of individuals with female and male gender identity orientations aligned with hetero/cis norms. Those with a male gender identity orientation were more likely to derive enjoyment from seducing and dominating their partners, considered sex important, and exhibited sexual attraction toward male attire, male-related activities, and seeing themselves as men. Conversely, individuals with an FGIO found pleasure in yielding themselves to their partners, indulging in clothing and other fetishes, and associating themselves with being women.

The respondents with a nonbinary GIO were less inclined to be exclusively attracted to binary genders (only men or women) and tended to exhibit a more bi- or panphilic pattern of sexual attraction. Many respondents in the present study reported being attracted to both men and women.

Although we found some adherence to traditional heterosexuality, the respondents were more likely to engage in sexual activities with women than with men. Across binary female, binary male, and nonbinary GIOs, many individuals expressed an interest in the gender-oriented fetishistic aspects of sexuality. In particular, those who leaned toward a nonbinary GIO were more inclined to have transgender individuals as their sexual partners. In terms of recent sexual activities, a minority reported engaging in vaginal or anal intercourse, while the majority mentioned activities like kissing, caressing, fingering/masturbation, and oral sex.

Notably, the fetishization of transgender and nonbinary individuals has been a subject of discussion concerning the objectification of trans bodies. This can manifest as an assumption that trans individuals undergo genital surgery to become more attractive to cisgender people or to be objectified in a fetishistic manner by cisgender individuals who are aroused by the transsexual body [62,63].

The most significant sexual challenge seems to be a lack of a suitable partner, with the majority currently living without a partner.

The present study has several limitations. First—and most significantly—it was a cross-sectional study, meaning that no causal inferences can be made based on the results. Furthermore, the data are descriptive and should not be interpreted extensively.

Second, the present study’s findings are susceptible to potential biases, including nonresponse bias and recall bias, because of the method of participant recruitment. Third, the size of the TGD population remains unknown, and having a response rate is essential for accurate calculations. However, even with these limitations, we found that we had a very good response rate. Finally, the present study was carried out in Norway, primarily with Caucasian participants, and the results may not apply to other cultural contexts or regions. However, the strength of the current study lies in its potential, because of its belonging to a liberal and relatively sex-positive society, to shed light on a topic that has been inadequately explored in the literature.

## 5. Conclusions

Our findings demonstrate that transgender individuals are actively engaged in sexuality and participate in a variety of sexual activities. However, a significant majority expressed dissatisfaction with their sexual lives, with their most common challenge being the absence of a suitable partner. By shifting the focus toward the pleasurable aspects of sexuality, the potential for experiencing genital pleasure from what might be considered “wrong” sex organs is evident. This phenomenon was particularly noticeable in a group of individuals who identified with a nonbinary gender orientation.

As researchers, we acknowledge that these results represent a step forward in a process leading to more acceptance and acknowledgment of sexual rights and pleasure among TGD individuals. One of the most prominent changes occurring in society is the growing recognition of trans identities as an enduring component of human diversity. We believe that, in trans communities, both femininity and masculinity are copied and deconstructed; so, trans communities may be somewhat avant-garde as compared to the rest of the community. This is not what we examined, but it could be explored further in another study.

## Figures and Tables

**Table 1 healthcare-12-00482-t001:** Demographics of the respondents answering the survey on sexuality and gender identity orientation (GIO).

	All		Female Gender Identity Orientation		Male Gender Identity Orientation		Nonbinary Gender Identity Orientation		
	n	(%)	n	(%)	n	(%)	n	(%)	*p*-Value
Age, years, n (%)									
<20	40	(12.9)	3	(3.4)	31	(20.7)	6	(8.3)	
21–30	155	(49.8)	43	(48.3)	86	(57.3)	26	(36.1)	
31–40	53	(17.0)	19	(21.3)	21	(14.0)	13	(18.1)	
41–50	35	(11.3)	15	(16.9)	10	(6.7)	10	(13.9)	
>50	28	(9.0)	9	(10.1)	2	(1.3)	17	(23.6)	<0.001
Unmarried or divorced (Marital status), n (%)	266	(85.0)	74	(83.1)	137	(92.7)	53	(71.6)	<0.001
Education, n (%)									
Primary	50	(16.2)	8	(9.1)	31	(20.9)	11	(15.1)	
Secondary	122	(39.5)	39	(44.3)	65	(43.9)	18	(24.7)	
Higher education	137	(44.3)	41	(46.6)	52	(35.1)	44	(60.3)	0.002
Employed, yes, n (%)	138	(44.1)	42	(47.2)	60	(40.0)	36	(48.6)	0.370

**Table 2 healthcare-12-00482-t002:** Sexual satisfaction by gender identity orientation (GIO).

			Gender Identity Orientation						
	All		Female Gender Identity Orientation		Male Gender Identity Orientation		Nonbinary Gender Identity Orientation		
	n	(%)	n	(%)	n	(%)	n	(%)	*p*-Value
What gives you sexual satisfaction? ^1^									
The pleasure of seducing	131	(41.9)	30	(33.7)	76	(50.7)	25	(33.8)	0.010
Giving yourself to someone	161	(51.4)	55	(61.8)	69	(46.0)	37	(50.0)	0.059
The pleasure of feeling another’s body	227	(72.5)	68	(76.4)	104	(69.3)	55	(74.3)	0.459
Notice sexual arousal	219	(70.0)	59	(66.3)	111	(74.0)	49	(66.2)	0.328
The pleasure of dominating the partner	71	(22.7)	9	(10.1)	45	(30.0)	17	(23.0)	0.002
Feeling of being affirmed as gender	155	(49.5)	46	(51.7)	75	(50.0)	34	(45.9)	0.756
Use of clothing and other fetishes	71	(22.7)	25	(28.1)	19	(12.7)	27	(36.5)	<0.001
How important is sexuality for you to feel good? ^2^	221	(70.6)	53	(59.6)	114	(76.0)	54	(73.0)	0.033
How satisfied are you with your sex life? ^3^	136	(43.5)	34	(38.2)	69	(46.0)	33	(44.6)	0.683

^1^ Responses: “Yes” and “No”. Proportion of respondents who answered “Yes”. ^2^ Responses: “Absolutely necessary”, “Important”, “Not very important”, “Not important at all”, “Don’t know”. Proportion of respondents who answered “Absolutely necessary” or “Important”. ^3^ Responses: “Very Satisfied”, “Satisfied”, “Somewhat dissatisfied”, “Not satisfied at all”, “Not relevant”. Proportion of respondents who answered “Very satisfied” or “Satisfied”.

**Table 3 healthcare-12-00482-t003:** Mean (SD) age of first sexual encounter by gender identity orientation.

			Gender Identity Orientation						
	All		Female Gender Identity Orientation		Male Gender Identity Orientation		Nonbinary Gender Identity Orientation		
	Mean	(SD)	Mean	(SD)	Mean	(SD)	Mean	(SD)	*p*-Value
Age (years) at first sex with oneself	12.7	(3.9)	13.1	(3.2)	12.8	(4.0)	12.8	(4.0)	0.823
Age (years) at first sex with another	17.5	(4.2)	18.5	(5.3)	16.8	(3.5)	17.8	(3.4)	0.015

**Table 4 healthcare-12-00482-t004:** Sexual attraction by gender orientation.

			Gender Identity Orientation						
	All		Female Gender Identity Orientation		Male Gender Identity Orientation		Nonbinary Gender identity Orientation		
	n	(%)	n	(%)	n	(%)	n	(%)	*p*-Value
Who do you feel sexually attracted to? ^1^									
Only women	65	(19.8)	19	(20.4)	36	(24.3)	10	(11.4)	0.034
Only men	28	(8.5)	10	(10.8)	15	(10.1)	3	(3.4)	0.106
Both men and women	122	(37.1)	38	(40.9)	65	(43.9)	19	(21.6)	0.001
Preferably transpersons	5	(1.5)	1	(1.1)	1	(0.7)	3	(3.4)	0.263
Personal characteristics regardless of gender	83	(25.2)	18	(19.4)	27	(18.2)	38	(43.2)	<0.001
Other	26	(7.9)	7	(7.5)	4	(2.7)	15	(17.0)	0.001

^1^ Responses: “Select the one that applies”.

**Table 5 healthcare-12-00482-t005:** Sexual turn-on patterns by gender identity orientation.

			Gender Identity Orientation						
	All		Female Gender Identity Orientation		Male Gender Identity Orientation		Nonbinary Gender Identity Orientation		
	n	(%)	n	(%)	n	(%)	n	(%)	*p*-Value
^1^ To what extent are you sexually attracted to:									
Female clothing	52	(16.0)	12	(13.0)	22	(15.4)	18	(19.8)	0.447
Male clothing	20	(6.3)	2	(2.2)	15	(10.7)	3	(3.4)	0.014
Female activities	31	(9.6)	13	(14.3)	10	(7.1)	8	(8.9)	0.186
Male activities	24	(7.4)	3	(3.2)	17	(12.0)	4	(4.4)	0.019
See myself as a woman	39	(12.1)	22	(23.9)	3	(2.1)	14	(15.7)	<0.001
See myself as a man	53	(16.3)	3	(3.3)	40	(28.0)	10	(11.0)	<0.001
See myself as trans	30	(9.2)	3	(3.3)	6	(4.3)	21	(22.8)	<0.001
Fetishes	81	(24.8)	27	(29.3)	31	(21.7)	23	(25.3)	0.412

^1^ Responses: “To a very large extent”, “To a large extent”, “Somewhat”, “To a small extent”, “To a very small extent”. “Not at all”. Proportion of respondents who answered “To a very large extent” or “To a large extent”.

**Table 6 healthcare-12-00482-t006:** Last sexual encounter by gender identity orientation.

			Gender Identity Orientation						
	All		Female Gender Identity Orientation		Male Gender Identity Orientation		Nonbinary Gender Identity Orientation		*p*-Value
	n	(%)	n	(%)	n	(%)	n	(%)	
The last person I had sex with was:									
Woman	142	(53.6)	47	(59.5)	63	(56.3)	32	(43.2)	0.100
Man	90	(34.0)	26	(32.9)	37	(33.0)	27	(36.5)	0.864
Transperson/intersex	33	(12.5)	6	(7.6)	12	(10.7)	15	(20.3)	0.046

**Table 7 healthcare-12-00482-t007:** Sexual practices by gender identity orientation.

			Gender Identity Orientation						
	All		Female Gender Identity Orientation		Male Gender Identity Orientation		Nonbinary Gender Identity Orientation		
	n	(%)	n	(%)	n	(%)	n	(%)	*p*-Value
^1^ What activities did you do at least once during the last time you were together?									
Kissing	184	(68.7)	66	(83.5)	66	(58.4)	52	(68.4)	0.001
Fingering/masturbation	176	(65.9)	58	(73.4)	66	(58.9)	52	(68.4)	0.099
Vaginal penetration	83	(30.7)	35	(44.3)	20	(17.5)	28	(36.4)	<0.001
Oral sex	124	(46.1)	50	(63.3)	41	(36.0)	33	(42.4)	<0.001
Anal penetration	45	(16.7)	22	(27.8)	12	(10.6)	11	(14.3)	0.006
Use of dildo	42	(15.6)	16	(20.3)	18	(15.8)	8	(10.4)	0.235
Number of activities (range 0–6), mean (SD)	2.4	(1.7)	3.1	(1.5)	2.0	(1.7)	2.4	(1.7)	<0.001
Achieved orgasm myself ^1^	111	(54.7)	37	(52.9)	40	(52.6)	34	(59.6)	0.247
The partner achieved orgasm ^1^	145	(71.1)	51	(72.9)	49	(64.5)	45	(77.6)	0.519
How often have you masturbated in the last 12 months?									
Often/very often	95	(45.9)	13	(18.6)	52	(67.5)	30	(50.0)	
Never/seldom/sometimes	112	(54.1)	57	(81.4)	25	(32.5)	30	(50.0)	<0.001

^1^ Responses: “Yes” and “No”. Proportion of respondents who answered “Yes”.

## Data Availability

Data are currently undergoing analyses for other research purposes and, thus, are not currently available.

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
