# Peer review of "Gender Identity Orientation and Sexual Activity—A Survey among Transgender and Gender-Diverse (TGD) Individuals in Norway"

_healthcare, 2024, doi:10.3390/healthcare12040482_

Round 1
Reviewer 1 Report
Comments and Suggestions for Authors
This is an article about the relationship between gender identity orientation and sexual activity. This is a relevant topic that the material for the article could serve as a starting point for investigating, but treating the material as an end point, with a presentation of undiscussed findings, does not realize this potential. The article starts with a summary presentation of shifts in thinking about what sex/gender means spanning almost 200 years, and what is mentioned seems a bit haphazard. The whole of the introduction should be revised to offer a briefer and more clearly framed presentation of relevant ideas and practices regarding sex/gender identity. I am also not sure that a presentation of the longer history is really needed for this particular article, especially as the presentation is quite sweeping. The John/Joan case is relevant, but too little detail is now included, so it is difficult for the reader to understand the nuances of the case. Also, the authors simplify complexities when they conclude that the re-operation was a return to ‘the gender he genuinely identified with’. After presenting the John/Joan case, the authors go on to present the role of medical advancements made, and the relationship between this and people’s identities is mentioned but could deserve some more attention.
The next section, ‘Current status and aims’, takes up some of the same issues (shifts in thinking about gender identity, diversity within sex/gender etc.) as the introduction, and I suggest that the authors restructure to make sure that 1.1 and 1.2 are assigned different tasks. The authors note that there has been a steep increase in research on trans* and this is an interesting point. What is presented of this body of research in 1.2 seems, again, quite haphazard and is also not clearly useful for the later presentation of findings. Here the authors also present their analytical framework, consisting of a combination of van Anders’ and Fausto-Sterling’s thinking, something I think works and is explained well. This framework does not really appear again later in the article, and I think the authors should explain what this framework has meant for their approach.
The methods section describes how the data has been produced but several questions remain. As gender and sexuality are highly diverse, and partially culturally contingent phenomenon, I would have liked to know what the authors mean when they state that the questionnaire has been ‘culturally adapted’ to suit a Norwegian context. Gendered norms and norms pertaining to sexuality are very different across the contexts where the questionnaire has already been applied. And when later in the article reading that being submissive or passive is coded as feminine, I cannot help to think that this is a stereotype more than a reality, at least in a context like Norway. While the authors do not proclaim that the data are representative, I do think recruitment strategies warrant some discussions. Activists and people in active treatment are very particular groups to include in a study like this. Also, while the practical work of re-classifying self-descriptions from the questionnaire is described at length, the purpose and consequences is not really described and discussed. What was the purpose of rating answers in terms of ‘femaleness’, ‘maleness’ and ‘binaryness’? Why was the group of raters selected to best be able to ascertain answers in these terms? Also, that clusters were found (page 6, line 275-276) in the results of a classification aimed at making clusters (grouping answers in terms of ‘femaleness’, ‘maleness’ and ‘binaryness’) is not surprising, but a result of the approach. This should not be presented as a process of ‘finding’ gender, but ‘making’ gender based on a set of ideas in the group of graders and the premises of a questionnaire developed elsewhere.
The main problem is not the weak points described above in terms of the execution of the article, but rather a lack of clarification of the value of the contribution. The aim of the study is somewhat unclear to me, or rather, the relevance and value of the aim is not clear. The authors (on page 5) state that the aim is to ‘investigate whether there are variations in sexual activities across different gender identity orientations.’. It is no surprise that they later find variations, but this aim does not offer anything in the way of an analysis of such variation, and the authors fail to explain why identifying such variation is of use to us. The authors are clear that the study is not representative, and the findings are presented as only a representation of the combination of gender identity orientation and sexual activity for the 336 people studied. Research can be relevant in a broader sense, not only by being statistically representative but also by offering analytical points that can be relevant beyond the study group. Towards the end of the article (in the Discussion) the aim of investigating ‘the sexual scripts of TGD individuals concerning their GIO’ is mentioned. This is a better aim of the article, but it is not clear how the present article could be revised to reach this aim instead. The data regarding sexual preferences are too thin to offer an understanding of such scripts. The Discussion does altogether not clearly follow from the presentation of the data. Instead, it contains different things, such as a programmatic statement that research on trans* should not be conducted without the involvement of trans people, without the article addressing the more complicated questions about how to deal with representation issues. Including someone because they are trans* with a view to have them represent the group, easily simplifies the question, conflates differences and thus essentialized the experience of being trans. Also, the Discussion includes the point that cis people objectify trans people. As the article is not on the sexual desires of cis people, I do not really see how this I relevant, unless the authors think that this is reflected in the sexual scripts of the respondents, and in that case, this should be motivated.
Details:
I do not really understand the presentation of the combinations identified by Hirschfeld presented on page 1. That these four groups of variations result in an enormous amount of combinations is a theoretical claim, not an empirical finding, and I don’t think the authors need this in order to establish that there is a great diversity of identities among humans.
The question of the relationship between ‘sex’ and ‘gender’ is of course difficult, but I think the authors should spell out their understanding early on and let the language of the rest of the article reflect this. The authors in many instances write ‘sex and gender’, also where I suspect they only mean one of these.
Page 2, lines 42-44: the first sentence on the page is programmatic, more than clarifying. What is meant by the statement: ‘the acknowledgment of individuals as existential beings that strengthened the recognition of individual rights, freedom, and respect’?
Page 2, line 52: when the authors write ‘this phenomenon’ it is not clear what this is in reference to.
Page 2, line 54: when the authors write ‘at that time’ it is not clear what time this is in reference to.
Page 2, line 59: the authors mention the ‘one-sex model’, this should be explained.
Page 2, line 67: language
Page 4, line 195: when making a global claim that there has been a ‘noticeable increase’ the authors should reference more than a one-country study with old data.
Page 10, line 356-358: the authors are clear on the fact that their data are not representative, and they should therefore not infer a statistical comparison between respondents and the general population in terms of satisfaction of sexual lives. When doing that they indicate that trans people are less satisfied with this than the general population, and they have no sound basis for making such a claim.
Comments on the Quality of English LanguageThe language is sufficiently clear. There are some typos and the article should be edited to correct these.
Author Response
Response to Reviewer 1
Reviewer 1, Comment 1
The article starts with a summary presentation of shifts in thinking about what sex/gender means spanning almost 200 years, and what is mentioned seems a bit haphazard. The whole of the introduction should be revised to offer a briefer and more clearly framed presentation of relevant ideas and practices regarding sex/gender identity. I am also not sure that a presentation of the longer history is really needed for this particular article, especially as the presentation is quite sweeping.
Authors response to comment 1 from reviewer 1
We have deleted some of the historical material and focused on the situation for transpeople where depathologizing has opened up for more pride and more sexual activity.
Reviewer 1, Comment 2
The John/Joan case is relevant, but too little detail is now included, so it is difficult for the reader to understand the nuances of the case. Also, the authors simplify complexities when they conclude that the re-operation was a return to ‘the gender he genuinely identified with’. After presenting the John/Joan case, the authors go on to present the role of medical advancements made, and the relationship between this and people’s identities is mentioned but could deserve some more attention.
Authors response to comment 2 from reviewer 1
We do not think that we need to describe the John/Joan case in detail, but we have deleted “genuinely” and write instead “the gender he identified with”. We have met Keith Sigmundsson and Milton Diamond who described the situation where David Reimer was informed about his history, and I think the description about David´s response is in accordance with how it was. The most important for the rest of us, is that the story caused a lot of reaction in the professional milieu, and eventually resulted in a new, less pathologizing understanding of gender incongruence.
Concerning the relationship between the role of medical advancement and people´s identities, we were in doubt whether this should be included – we decided to include it “in passing”, but not elaborate on it, since these ideas is more part of history that the present. This might be of interest in another paper.
Reviewer 1, Comment 3
The next section, ‘Current status and aims’, takes up some of the same issues (shifts in thinking about gender identity, diversity within sex/gender etc.) as the introduction, and I suggest that the authors restructure to make sure that 1.1 and 1.2 are assigned different tasks.
Authors response to comment 3 from reviewer 1
We have reorganized 1.1 Historical background and 1.2 Gender identity orientations among trans and gender diverse people and introduced a new 1.3. Sexuality among trans and gender diverse people.
Reviewer 1, Comment 4
The authors note that there has been a steep increase in research on trans* and this is an interesting point. What is presented of this body of research in 1.2 seems, again, quite haphazard and is also not clearly useful for the later presentation of findings.
Authors response to comment 4 from reviewer 1
We agree with the reviewer that this section may not be directly relevant to the presentation and discussion of findings. It was intended as background information and a reasoning for the need of more research on non-binary gender identity orientations.
We have changed the following text:
This surge in publications may reflect a growing societal acceptance and a professional interest in gender identity orientations within the transgender community.
Into:
This surge in publications reflect a growing professional interest in gender identity orientations within the transgender community, especially into non-binary gender identity orientations.
Reviewer 1, Comment 5
Here the authors also present their analytical framework, consisting of a combination of van Anders’ and Fausto-Sterling’s thinking, something I think works and is explained well. This framework does not really appear again later in the article, and I think the authors should explain what this framework has meant for their approach.
Authors response to comment 5 from reviewer 1
Van Anders and Fausto-Sterling has been included the discussion:
From the self-descriptions provided by the respondents, we have identified trans-identities as differentiated, better understood as dimensions related to degree of maleness, femaleness and non-binaryness rather than polarizations, in line with Sandra Bems and Anne Fausto-Sterlings suggestions. As a new development we observe more bisexuality and sexual activities that can be further described as configurations as sug-gested by Sari van Anders.
Reviewer 1, Comment 6
The methods section describes how the data has been produced but several questions remain. As gender and sexuality are highly diverse, and partially culturally contingent phenomenon, I would have liked to know what the authors mean when they state that the questionnaire has been ‘culturally adapted’ to suit a Norwegian context.
Authors response to comment 6 from reviewer 1
We thank the reviewer for this comment and have changed the text as follows:
The French questionnaire (Giami & Beaubatie, 2014; Giami et al., 2011) was translated utilizing the forward-backward translation procedure (Maneesriwongul & Dixon, 2004) where the French questionnaire were translated to Norwegian by a bilingual translator. Cultural differences between Norway and France were discussed in the Norwegian research team and necessary changes (eg. Educational and Health care systems) in the questionnaire were done. Lastly, the questionnaire was back-translated to French and the French version was reviewed by the French authors.
Reviewer 1, Comment 7
Gendered norms and norms pertaining to sexuality are very different across the contexts where the questionnaire has already been applied. And when later in the article reading that being submissive or passive is coded as feminine, I cannot help to think that this is a stereotype more than a reality, at least in a context like Norway.
Authors response to comment 7 from reviewer 1
We do not think we have used the words ‘submissive’ or ‘passive’ in our text.
Reviewer 1, Comment 8
While the authors do not proclaim that the data are representative, I do think recruitment strategies warrant some discussions. Activists and people in active treatment are very particular groups to include in a study like this. Also, while the practical work of re-classifying self-descriptions from the questionnaire is described at length, the purpose and consequences is not really described and discussed. What was the purpose of rating answers in terms of ‘femaleness’, ‘maleness’ and ‘binaryness’?
Authors response to comment 8 from reviewer 1
Usually, in surveys, respondents are asked to check out predefined gender categories. As stated in the manuscript (Methods section) our respondents were asked to describe their gender identity in their own words. This method gives very rich data that can be analyzed in different ways. In this paper, the written responses were classified along three dimensions, degree of feminine, masculine or non-binary gender identity orientation.
We agree with the reviewer and have moved the following sections to a more prominent place in the manuscript:
‘Gender expressions can be viewed along a continuum, ranging from binary to non-binary, drawing inspiration from Sandra Bem's research on gender roles. Originally, Bem conceptualized femininity and masculinity as opposing ends of a single dimension. However, when she introduced the idea of masculinity and femininity as two independent dimensions, it opened up new possibilities, allowing individuals to embrace both masculine and feminine qualities and neither simultaneously (Bem, 1974; Bem SL, 1993).
The concept of a non-binary notion of transgenderism appears to be relatively recent. A search conducted in library databases[1], including Medline, Cinahl, SocIndex, Teacher Reference Center, Health and Psychosocial Instruments, and APA PsychInfo, using the terms ‘non-binary’ or ‘nonbinary’ in the title and ‘gender’ in the abstract, revealed 2 peer-reviewed publications before 2010, with a sharp increase from 2015 (n=5), to 2020 (n=120), until end 2022 (n=282). This surge in publications may reflect a growing professional interest in gender identity orientations within the transgender community, especially into non-binary gender identity orientations.
In sum we believe gender identity orientation can be viewed as TGD individuals being placed in three different dimensions. These dimensions encompass the degree of maleness, distinct from the degree of femaleness, and distinct from the degree of non-binarity. While numerous challenges persist for TGD individuals, progress has been made toward reducing pathologization and increasing acceptance, creating a self-defined space where one can embrace both their gender identity and their sexuality.’
Reviewer 1, Comment 9
Why was the group of raters selected to best be able to ascertain answers in these terms? Also, that clusters were found (page 6, line 275-276) in the results of a classification aimed at making clusters (grouping answers in terms of ‘femaleness’, ‘maleness’ and ‘binaryness’) is not surprising, but a result of the approach. This should not be presented as a process of ‘finding’ gender, but ‘making’ gender based on a set of ideas in the group of graders and the premises of a questionnaire developed elsewhere.
Authors response to comment 9 from reviewer 1
The group of raters consisted of 10 experienced clinicians and 10 TGD individuals. We believed that having an external group of raters would give a more valid rating.
Reviewer 1, Comment 10
The main problem is not the weak points described above in terms of the execution of the article, but rather a lack of clarification of the value of the contribution. The aim of the study is somewhat unclear to me, or rather, the relevance and value of the aim is not clear. The authors (on page 5) state that the aim is to ‘investigate whether there are variations in sexual activities across different gender identity orientations.’. It is no surprise that they later find variations, but this aim does not offer anything in the way of an analysis of such variation, and the authors fail to explain why identifying such variation is of use to us.
Authors response to comment 10 from reviewer 1
We thank the reviewer for this comment. We have clarified this point by refering to the new freedom to be openly sexual after depathologizing, and probably also as a result of cultural acceptance. We have also taken into account that this is a new field of research and not an “end point”.
We have specified the aim as follows:
‘This study aims to investigate variations in sexual activities across different gender identity orientations.’
Reviewer 1, Comment 11
The authors are clear that the study is not representative, and the findings are presented as only a representation of the combination of gender identity orientation and sexual activity for the 336 people studied. Research can be relevant in a broader sense, not only by being statistically representative but also by offering analytical points that can be relevant beyond the study group. Towards the end of the article (in the Discussion) the aim of investigating ‘the sexual scripts of TGD individuals concerning their GIO’ is mentioned. This is a better aim of the article, but it is not clear how the present article could be revised to reach this aim instead.
Authors response to comment 11 from reviewer 1
See above. “Sexual scripts” can be discussed, but if we refer to the sexual script theory by Simon and Gagnon, what we observe is remnant from traditional heteronormative scripts, but also sexual activities without a template or script, as new possibilities and new identities develop.
Reviewer 1, Comment 12
The data regarding sexual preferences are too thin to offer an understanding of such scripts.
Authors response to comment 12 from reviewer 1
The data are new and describing a budding process and should therefore not be interpreted to extensively.
We have thus added this into the strength/weakness section:
‘This study presents several limitations. Firstly, and most significantly, it was a cross-sectional study, which means that no causal inferences can be made based on the results. Furthermore, the data are descriptive and should therefore not be interpreted to extensively.’
Reviewer 1, Comment 13
The Discussion does altogether not clearly follow from the presentation of the data. Instead, it contains different things, such as a programmatic statement that research on trans* should not be conducted without the involvement of trans people, without the article addressing the more complicated questions about how to deal with representation issues. Including someone because they are trans* with a view to have them represent the group, easily simplifies the question, conflates differences and thus essentialized the experience of being trans.
Authors response to comment 13 from reviewer 1
We have removed this paragraph.
Reviewer 1, Comment 14
Also, the Discussion includes the point that cis people objectify trans people. As the article is not on the sexual desires of cis people, I do not really see how this I relevant, unless the authors think that this is reflected in the sexual scripts of the respondents, and in that case, this should be motivated.
Authors response to comment 14 from reviewer 1
The point about objectifying and fetishizing of transpeople has been included because it may affect transpeople´s idea about fulfilling cis-normative sexual ideals.
Reviewer 1, Comment 15
I do not really understand the presentation of the combinations identified by Hirschfeld presented on page 1. That these four groups of variations result in an enormous amount of combinations is a theoretical claim, not an empirical finding, and I don’t think the authors need this in order to establish that there is a great diversity of identities among humans.
Authors response to comment 15 from reviewer 1
Hirschfeld is taken out, not because it is a theoretical claim – as such it is of interest that this was a way of thinking. It is taken out because we have focused more on issues that show the development from pathologizing to normalization of gender diversity.
Reviewer 1, Comment 16
The question of the relationship between ‘sex’ and ‘gender’ is of course difficult, but I think the authors should spell out their understanding early on and let the language of the rest of the article reflect this. The authors in many instances write ‘sex and gender’, also where I suspect they only mean one of these.
Authors response to comment 16 from reviewer 1
We have rewritten all passages with ‘sex and gender’ to eliminate such misunderstandings.
Reviewer 1, Comment 17
Page 2, lines 42-44: the first sentence on the page is programmatic, more than clarifying. What is meant by the statement: ‘the acknowledgment of individuals as existential beings that strengthened the recognition of individual rights, freedom, and respect’?
Authors response to comment 17 from reviewer 1
We agree with the reviewer and have deleted this sentence from the manuscript.
Reviewer 1, Comment 18
Page 2, line 52: when the authors write ‘this phenomenon’ it is not clear what this is in reference to.
Authors response to comment 18 from reviewer 1
“phenomenon” has been replaced by “gender diversity”.
Reviewer 1, Comment 19
Page 2, line 54: when the authors write ‘at that time’ it is not clear what time this is in reference to.
Authors response to comment 19 from reviewer 1
“At that time” is taken out.
Reviewer 1, Comment 20
Page 2, line 59: the authors mention the ‘one-sex model’, this should be explained.
Authors response to comment 20 from reviewer 1
To mention that there has been a one-sex model is of importance since the two-sex model is av relatively recent origin in history. Here it is also of importance that sex and gender has been merged for a long time in history, where the understanding of “man” and “woman” was based on the sex organs and the capacity to procreate. The “one sex mode”l was based on the idea that a man was a more developed creature than a woman, and women did for centuries not regarded as being equipped with a soul, unlike men. I did not think this was important to explain?
Reviewer 1, Comment 21
Page 2, line 67: language
Authors response to comment 21 from reviewer 1
Refers to the English language? Changed to “In English…”
Reviewer 1, Comment 22
Page 4, line 195: when making a global claim that there has been a ‘noticeable increase’ the authors should reference more than a one-country study with old data.
Authors response to comment 22 from reviewer 1
We agree with the reviewer and have added the following reference: Study from New Zealand (Clarke), from Brazil (Spirizzi) and a literature review (Zhou)
Reviewer 1, Comment 23
Page 10, line 356-358: the authors are clear on the fact that their data are not representative, and they should therefore not infer a statistical comparison between respondents and the general population in terms of satisfaction of sexual lives. When doing that they indicate that trans people are less satisfied with this than the general population, and they have no sound basis for making such a claim.
Authors response to comment 23 from reviewer 1
We agree with the reviewer and have deleted the sentence in question.
Reviewer 2 Report
Comments and Suggestions for Authors
This paper seems to belong to a constellation of contributions relating to this phenomenon, a phenomenon that has only recently emerged in the scholarly literature. In a sense, we can only expect nuances (as opposed to writings that considerably advance the field) in any particular scholarly paper; the present draft here certainly qualifies as an example of this tendency.
We only have a few suggestions for improvement to improve the article's suitability for the intended audience. In the first place, the historical background section (lines 28~116) seems disconnected from the main portion of the paper. From the beginning of the section, the authors ought to establish the relevance of the historical background to the thesis of the article.
Another point that needs emphasis rests in a claim from the abstract in which the authors describe the reality of how they adapted a questionnaire in France to the Norwegian context (line 17). We find that portion of the abstract mentioned again in lines 218 to 220 ("The French questionnaire [49,50] was translated and culturally adapted to fit the Norwegian context, utilizing the forward-backward translation procedure (55). On the other hand, we never really hear anything further about the particularities through which the adaptation took place. The authors seem to presume that the audience already knows about the so-called forward-backward translation procedure (line 219), but the authors would not hurt the article by further expounding the intricacies of how the adaptation process took place.
Along these same lines, the authors frequently speak about the study as an event that unfolded in the Norwegian context (i.e., lines 395~396), but we never really learn about the specific intricacies of what the Norwegian context actually means and how that context would differ from, say, other European or non-European contexts.
In the discussion, the authors mention the following idea: "From the self-descriptions provided by the participants, we have identified the need to view trans-identities as differentiated (340~341)." On the other hand, the authors might want to elaborate on what this point actually means by highlighting relevant findings in the scholarly literature; lines 339 through 352 lack those kinds of references.
In this same paragraph of lines 339 through 352, the authors also add the following point: "Research is of paramount importance, and ideally, it should be conducted by trans-individuals themselves, with the support of allies who are engaged in the transgender community (348~350)." While the necessity for trans-individuals to serve as academic authors in this field might seem obvious to the authors, the authors ought to still explain the indispensability of trans-individuals as academic contributors. By way of analogy, nobody has ever said that Japanese people alone serve as the most ideal kinds of scholarly authors on issues relating to Japanese history; given that analogy, the claim of a field that needs trans-individuals as research study authors (lines 348~349) needs further explanation.
Author Response
Reviewer 2
Reviewer 2, Comment 1
We only have a few suggestions for improvement to improve the article's suitability for the intended audience. In the first place, the historical background section (lines 28~116) seems disconnected from the main portion of the paper. From the beginning of the section, the authors ought to establish the relevance of the historical background to the thesis of the article.
Authors response to comment 1 from reviewer 2
Thank you for your suggestion, the section has been revised.
Reviewer 2, Comment 2
Another point that needs emphasis rests in a claim from the abstract in which the authors describe the reality of how they adapted a questionnaire in France to the Norwegian context (line 17). We find that portion of the abstract mentioned again in lines 218 to 220 ("The French questionnaire [49,50] was translated and culturally adapted to fit the Norwegian context, utilizing the forward-backward translation procedure (55). On the other hand, we never really hear anything further about the particularities through which the adaptation took place. The authors seem to presume that the audience already knows about the so-called forward-backward translation procedure (line 219), but the authors would not hurt the article by further expounding the intricacies of how the adaptation process took place.
Authors response to comment 2 from reviewer 2
We thank the reviewer for this comment and have changed the text as follows:
The French questionnaire (Giami & Beaubatie, 2014; Giami et al., 2011) was translated utilizing the forward-backward translation procedure (Maneesriwongul & Dixon, 2004) where the French questionnaire were translated to Norwegian by a bilingual translator. Cultural differences between Norway and France were discussed in the Norwegian research team and necessary changes (eg. Educational and Health care systems) in the questionnaire were done. Lastly, the questionnaire was back-translated to French and the French version was reviewed by the French authors.
Reviewer 2, Comment 3
Along these same lines, the authors frequently speak about the study as an event that unfolded in the Norwegian context (i.e., lines 395~396), but we never really learn about the specific intricacies of what the Norwegian context actually means and how that context would differ from, say, other European or non-European contexts.
Authors response to comment 3 from reviewer 2
I agree with the reviewer. We have therefore given a short description of the Norwegian context at the end of the Discussion: “However, the strength of this study lies in its potential due to belonging to a liberal and relatively sex-positive society, to shed light on a topic that has been inadequately explored in existing research literature.”
Reviewer 2, Comment 4
In the discussion, the authors mention the following idea: "From the self-descriptions provided by the participants, we have identified the need to view trans-identities as differentiated (340~341)." On the other hand, the authors might want to elaborate on what this point actually means by highlighting relevant findings in the scholarly literature; lines 339 through 352 lack those kinds of references.
Authors response to comment 4 from reviewer 2
We have references to gender identity in the Historical background (4-11)
Reviewer 2, Comment 5
In this same paragraph of lines 339 through 352, the authors also add the following point: "Research is of paramount importance, and ideally, it should be conducted by trans-individuals themselves, with the support of allies who are engaged in the transgender community (348~350)." While the necessity for trans-individuals to serve as academic authors in this field might seem obvious to the authors, the authors ought to still explain the indispensability of trans-individuals as academic contributors. By way of analogy, nobody has ever said that Japanese people alone serve as the most ideal kinds of scholarly authors on issues relating to Japanese history; given that analogy, the claim of a field that needs trans-individuals as research study authors (lines 348~349) needs further explanation.
Authors response to comment 5 from reviewer 2
We agree with the reviewer and have deleted the sentence in question.
Reviewer 3 Report
Comments and Suggestions for Authors
It would be of interest/improve the scientific significance of this paper if the authors noted: 1. how many of the respondents self-identified as not-Caucasians; 2. how higher education+age+non-binary (NBGIO) dimensions collapsed.
It would also be of interest to see a reference to current US/UK survey studies (such ax The Williams Institute of Law) on how the GenZ now self-identifies as "bi" in larger numbers (almost 1 of 5) than the elder generations as the dominant respondent age cohort in this study comes close to GenZ.
It is also of interest that it seems that trans-identifying people find more sexual partners among women than among men: perhaps a reference to recent survey studies on bisexuality (which show that more women and more GenZ identify as bi than in other categories).
In any case, this is an important piece and I see no problems to publish it in its current form. My comments could be seen as important for further data analyses.
Author Response
Response to Reviewer 3:
Reviewer 3, Comment 1
It would be of interest/improve the scientific significance of this paper if the authors noted: 1. how many of the respondents self-identified as non-Caucasians;
Authors response to comment 1 from reviewer 3
We point to our statement in the strength/weakness section:
‘Lastly, this study was carried out in Norway, primarily with Caucasian participants, and the results may not apply to other cultural contexts or regions.’
Reviewer 3, Comment 2
- how higher education+age+non-binary (NBGIO) dimensions collapsed.
Authors response to comment 2 from reviewer 3
We agree with the reviewer that this is an interesting question. We indeed found that the majority of NBGIOs were younger persons born as females. However, this is a bit on the side of the paper’s aim, and we chose not to publish this result.
Reviewer 3, Comment 3
It would also be of interest to see a reference to current US/UK survey studies (such ax The Williams Institute of Law) on how the GenZ now self-identifies as "bi" in larger numbers (almost 1 of 5) than the elder generations as the dominant respondent age cohort in this study comes close to GenZ.
Authors response to comment 3 from reviewer 3
We certainly agree with the reviewer that the current surveys from the US (eg. Gallup Poll Social Series) are interesting. However, our primary objective was to describe and investigate sexual patterns among three trans dimensions (FGIO, MGIO and NBGIO). The very interesting new data on NBGIOs in the US may help to understand why a relatively large proportion of these are sexually attracted to a wide array of genders. But this falls a bit out of the intended aim of the study.
Reviewer 3, Comment 4
It is also of interest that it seems that trans-identifying people find more sexual partners among women than among men: perhaps a reference to recent survey studies on bisexuality (which show that more women and more GenZ identify as bi than in other categories).
Authors response to comment 4 from reviewer 3
Same as comment 3
Round 2
Reviewer 1 Report
Comments and Suggestions for Authors
I have read the authors’ comments and the revised article. I find that is has greatly improved on several points. I find that the reformulation of the purpose of the article was very fruitful, and the exploratory character of the piece is clearer now. I would like to point out that the questions are more about sexual preferences than sexual activities, so I suggest the author revise the purpose further. The historical section is now much more to the point, and the point of the rating exercise is more clearly described and motivated now. The authors have injected some formulations that now more clearly convey what they can and what they cannot say based on the material.
I still find that the authors fail to address issues I think are substantial to the contribution, especially the lack of addressing the cultural variability of sexual desire and practice. As a sexualities scholar, I find the presentation of sexual norms without reference to context and broader gender norms in Norwegian society, problematic. That the authors do not really see this in the same perspective, is clear in the authors’ response to Comment 7. There I write: “And when later in the article reading that being submissive or passive is coded as feminine, I cannot help to think that this is a stereotype more than a reality, at least in a context like Norway“. Instead of responding to this, the authors stress that these were not the terms they applied. While that is correct, I find it unhelpful that the authors fail to acknowledge that I use the terms ‘submissive and passive’ to sum up what the authors describe as a preference for “Giving yourself to someone”/”yielding themselves to their partners”. The authors do not respond to the substance of this critique. The point of my comment on this is that the authors seem to apply a blanket interpretation of such desires as ‘feminine’ when they state that this is an area where individuals in their study with a female gender identity orientation align with hetero/cis norms when they have a preference for “yielding themselves to their partners”. By not referencing this claim about hetero/cis norms and by not qualifying this claim in relation to gender relations and sexuality norms in contemporary Norway, the authors appear to essentialize this as a trait in femininity. The strength of the association between such sexual preferences and femininity in relation to hetero/cis norms would probably be quite different in different countries. Similarly, the very generalised link made between male gender identity and “The pleasure of dominating the partner”, should be motivated.
In relation to the authors’ answer to my comment 14 about the desires of cis people, I have no problem with the claim that this is something that may affect the acts and ideas of trans people. At the same time, the authors just mention this in passing and the claim is not made relevant to the discussion of the material at hand. The relevance is therefore still not clear to me. The inclusion of this may also seem a bit misleading as the paragraph on this sits between two paragraphs that offer a presentation of findings from the study.
Just to clarify: when I in Comment 21 stated ‘language’ in reference to page 2, line 67, this was meant to indicate that there was a language error on that line. ‘Whay’ should probably be ‘What’.
Comments on the Quality of English Language
No major problems detected.
Author Response
Comment 1: I would like to point out that the questions are more about sexual preferences than sexual activities, so I suggest the author revise the purpose further.
The questions in this survey are the following:
- Age
- Education
- What gives you sexual satisfaction?
- Age of first sex with oneself – and another person
- Who do you feel attracted to?
- To what extent are you attracted to:
- The last person I had sex with was:
- What activities did you do at least once during tha last time you were together;
- How often have you masturbated the last 12 months.
We cannot agree that this is more about preferences than sexual activities, even if preferences are part of the questions, and do not think that the purpose of the research should be revised on this point.
Comment 2: the lack of addressing the cultural variability of sexual desire and practice. I find the presentation of sexual norms without reference to context and broader gender norms in Norwegian society, problematic. “And when later in the article reading that being submissive or passive is coded as feminine, I cannot help to think that this is a stereotype more than a reality, at least in a context like Norway“.
We think that the reviewer might have misunderstood the manuscript at this point: We did not “code” being submissive or dominant as feminine or masculine, we asked about “giving yourself to someone” and “dominating the partner”, and the results show that in the “Female gender identity orientation” group, 61,8% report sexual satisfaction by “Giving yourself to someone”, while 76,4% report sexual satisfaction by “the pleasure of feeling another´s body”, and 66,3% report sexual satisfaction by noticing sexual arousal. We do not see our results as stereotypical, rather as a mix of cultural sexual patterns. In the “Male gender identity orientation” group, 30% respond sexual satisfaction by the pleasure of dominating the partner, and 46% respond that they get sexual satisfaction by giving themselves to someone. In the non-binary group, the results show that 50% respond that they get sexual satisfaction by giving themselves to someone.
We see that these results could have been discussed further, but we want to leave them as an expression of the situation as it is here and now, in a relatively liberal and sex positive country, where the trans population has entered the sexual arena where the interact with other transpeople, cis-people, and we find that all groups seems to prefer women as sexual partners at this time.
I find it unhelpful that the authors fail to acknowledge that I use the terms ‘submissive and passive’ to sum up what the authors describe as a preference for “Giving yourself to someone”/”yielding themselves to their partners”.
We understand that “giving yourself to someone” can be interpreted as being “submissive and passive”, but we also see that giving yourself to someone may be a part of sexual interaction between two equally active or passive partners. For us it is important to keep the descriptions as open as possible, without what we regard as unnecessary and imposed normativity evaluations.
the authors seem to apply a blanket interpretation of such desires as ‘feminine’ when they state that this is an area where individuals in their study with a female gender identity orientation align with hetero/cis norms when they have a preference for “yielding themselves to their partners”
In the manuscript we write: Our findings indicate that, to some extent, the sexual preferences of individuals with female and male gender identity orientations align with hetero/cis-norms, then we continue to show some examples of how some of the results may indicate identification with hetero/cis-norms. (line 322-329)
We see that we could have discussed this further, but since none of the results are conclusive, and we regard them as a picture of a process that is on continuous development, we have decided not to interpret them any further.
By not referencing this claim about hetero/cis norms and by not qualifying this claim in relation to gender relations and sexuality norms in contemporary Norway, the authors appear to essentialize this as a trait in femininity.
This is in our view a strange conclusion, we think this is exactly the opposite of what we do. The reviewer claim that we have “coded” “submissiveness” as “feminine” - this seems to be a misreading of the tables and the text. This is an interesting discussion, however, but we see this as a sidestep in the discussion we have presented based on our reading of the results.
The strength of the association between such sexual preferences and femininity in relation to hetero/cis norms would probably be quite different in different countries.
Similarly, the very generalised link made between male gender identity and “The pleasure of dominating the partner”, should be motivated.
We understand the reveiwer´s comment as misreading of our results. What we have reported is as described above that the group who orient themselves towards a feminine gender identity, report a higher inclination to have sexual pleasure from giving themselves to someone. This may be interpreted as identification to the traditional hetero/cis-norms, but anyway we see this a a finding along two dimensions: Gender identity orientation and route to sexual satisfaction, and as such they should not be interpreted as categories, or elaborated in relation to cultural differences in this context.
Comment 3: In relation to the authors’ answer to my comment 14 about the desires of cis people, I have no problem with the claim that this is something that may affect the acts and ideas of trans people. At the same time, the authors just mention this in passing and the claim is not made relevant to the discussion of the material at hand. The relevance is therefore still not clear to me. The inclusion of this may also seem a bit misleading as the paragraph on this sits between two paragraphs that offer a presentation of findings from the study.
We do not make a big issue of the possibility that trans people may be influenced by the cis-culture, but it is worth mentioning, and we give several examples in the paragraph where it is mentioned in the discussion (Paragraph 3, line 322-329).
Comment 4: Just to clarify: when I in Comment 21 stated ‘language’ in reference to page 2, line 67, this was meant to indicate that there was a language error on that line. ‘Whay’ should probably be ‘What’.
Thank you, this has been corrected.
Thanks for your comments, we have been inspired to think deeper, and we see that you are raising an interesting discussion, that we regard relevant, but more as a general analysis of how femininity and masculinity are understood and practiced in different societies.
The following paragraph has been added to the manuscript:
We believe that in trans communities femininity and masculinity both are copied and deconstructed, and trans communities may therefore be somewhat avant garde of the rest of the community. This is not what we have examined but could be explored further in another study.
We can organize language editing for this manuscript to be submitted at February 8th.
Sincerely,